## [Decision Letter · Decision Letter 0]

2 Dec 2025

Coinfection with malaria alters the dynamics and fitness of an intestinal nematode

Dear Dr. Sorci,

Thank you for submitting your manuscript to PLOS Neglected Tropical Diseases. After careful consideration, we feel that it has merit but does not fully meet PLOS Neglected Tropical Diseases's publication criteria as it currently stands. Therefore, we invite you to submit a revised version of the manuscript that addresses the points raised during the review process.

Please submit your revised manuscript within by Jan 30 2026 11:59PM. If you will need more time than this to complete your revisions, please reply to this message or contact the journal office at plosntds@plos.org. Please include the following items when submitting your revised manuscript:

We look forward to receiving your revised manuscript.

Kind regards,

Chao Yan

Academic Editor

Gabriel Rinaldi

Section Editor

Shaden Kamhawi

co-Editor-in-Chief

Paul Brindley

co-Editor-in-Chief

**Journal Requirements:**

At this stage, the following Authors/Authors require contributions: Luc Bourbon, Aloïs Dusuel, Emma Groetz, Mickaël Rialland, Benjamin Roche, Bruno Faivre, and Gabriele Sorci. Please ensure that the full contributions of each author are acknowledged in the "Add/Edit/Remove Authors" section of our submission form.

- ® on pages: 8, 9, and 10

- TM on pages: 8, 9, and 10.

5) We have noticed that you have uploaded Supporting Information files, but you have not included a list of legends. Please add a full list of legends for your Supporting Information files after the references list.

6)  Should your submission be accepted, we will require the following information in your Data Availability Statement:

1. The DOI provided by Dryad

2. The citation for your data package in the reference section of your manuscript

3. The citation for your data package in the methods section

If you are unable to adhere to our open data policy, please kindly revise your statement to explain your reasoning and we will seek the editor's input on an exemption. Please be assured that, once you have provided your new statement, the assessment of your exemption will not hold up the peer review process.

7) Please revise your current Competing Interest statement to the standard "The authors have declared that no competing interests exist."

**Reviewers' Comments:**

Reviewer's Responses to Questions

**Key Review Criteria Required for Acceptance?**

**Methods**

-Are the objectives of the study clearly articulated with a clear testable hypothesis stated?

-Is the study design appropriate to address the stated objectives?

-Is the population clearly described and appropriate for the hypothesis being tested?

-Is the sample size sufficient to ensure adequate power to address the hypothesis being tested?

-Were correct statistical analysis used to support conclusions?

-Are there concerns about ethical or regulatory requirements being met?

Reviewer #1: Yes to all

Reviewer #2: There are some good strengths here. Experimental design is appropriate, with well-controlled infection-order comparisons (malaria before/after helminth). The IL-13 manipulation experiments (recombinant supplementation and neutralization) effectively establish causality between Th2 suppression and increased helminth fecundity. Thought I do have several concerns:

The statistical models (GLMMs) need full specification: distribution family, link function, and treatment of zero inflation in fecundity data.

Clarify biological vs. technical replication for all assays and indicate sample sizes in figure legends. If feasible, include intestinal-level immune data (e.g., IL-13 or IL-4 expression, goblet or tuft cell counts) to support the systemic cytokine findings. If not available, the limitation should be explicitly discussed.

I would also ask the authors to provide details on how post-hoc corrections (Bonferroni or otherwise) were applied.

Reviewer #3: The study’s objectives and hypotheses are clearly stated, and the coinfection experimental design, including use of C57BL/6 mice and the described infection protocols, is appropriate for the mechanistic questions posed. The study population and sample sizes appear suitable for detecting the main effects reported, even though formal power calculations are not provided. Overall, the statistical framework is appropriate, but the authors should briefly clarify how zero egg counts were handled and justify the use of Gaussian models rather than, for example, negative binomial GLMMs, or acknowledge this as a limitation. Ethical approvals and animal care procedures are adequately described and appear fully compliant with regulatory standards

**Results**

-Does the analysis presented match the analysis plan?

-Are the results clearly and completely presented?

-Are the figures (Tables, Images) of sufficient quality for clarity?

Reviewer #1: Yes to all

Reviewer #2: The results are well organized and clearly demonstrate that malaria coinfection enhances worm persistence and fecundity and the IL-13 manipulation experiments are convincing and mechanistically informative.

My concerns here include:

The term “parasite fitness” should be used cautiously—current measurements reflect population-level egg output and persistence rather than individual reproductive success. Consider rephrasing as “increased fecundity and persistence.”

The role of other Th2 cytokines (IL-4, STAT6 pathway) should be acknowledged, even if IL-13 is dominant.

Statistical figures should include explicit n values and variance representation (e.g., SEM, SD).

Reviewer #3: The analyses presented are consistent with the stated aims and appear to follow the analysis plan implied by the Methods. Results are clearly and systematically presented, with key outcomes (egg excretion, worm biomass, persistence, immune readouts) appropriately summarized in the text and figures. The figures are of generally good quality and sufficient for clarity.

**Conclusions**

-Are the conclusions supported by the data presented?

-Are the limitations of analysis clearly described?

-Do the authors discuss how these data can be helpful to advance our understanding of the topic under study?

-Is public health relevance addressed?

Reviewer #1: Yes to all

Reviewer #2: The conclusion that malaria infection enhances helminth reproduction via suppression of IL-13–mediated immunity is well supported. I also think that the study provides an elegant example of how immune polarization affects inter-parasite ecological interactions.

However, the discussion overstates implications for “parasite fitness” and transmission potential; temper these statements. I also think that the extrapolation to human malaria–helminth coinfections should be clearly framed as speculative. The conclusions would be improved by the addition of a short paragraph connecting these findings to public-health or epidemiological implications (e.g., how malaria control might indirectly influence STH transmission).

Reviewer #3: The conclusions are generally well supported by the data, and the authors clearly articulate how their findings advance understanding of helminth–malaria coinfection and parasite fitness. They do address limitations, but the discussion should more explicitly acknowledge the restricted generalizability of this murine H. polygyrus–Plasmodium model to human STH–malaria systems and the uncertainty in extrapolating quantitative effects to human epidemiology. The public health relevance, particularly implications for STH control and surveillance, is clearly highlighted, but would benefit from slightly tempered claims and clearer framing as mechanistic proof of concept, which will increase rigor of the manuscript.

**Editorial and Data Presentation Modifications?**

Reviewer #1: (No Response)

Reviewer #2: Minor grammatical and stylistic edits needed (e.g., “Coinfection can exacerbate disease severity” instead of “has the potential to worsen symptoms”). Please ensure all species names are italicized and cytokines consistently formatted (IL-13, IL-4, IFN-γ).

On the figures, add y-axis labels with units and indicate scale (log10 where applicable). Please explicitly state sample sizes in legends. Clarify whether lines represent mean ± SEM or SD.

I would also suggest that the authors include or reference access to raw numerical data per PLOS data policy.

Reviewer #3: Accept

**Summary and General Comments**

Reviewer #1: Review of the manuscript "Coinfection with malaria alters the dynamics and fitness of an intestinal nematode"

Thank you for giving me the opportunity to review this manuscript. In this study, the authors investigate the effects of coinfection with Plasmodium yoelii and Heligmosomoides polygyrus in mice. They report that coinfection leads to increased fecundity and fitness of H. polygyrus, likely due to a shift toward a Th1-skewed immune response and an attenuation of the Th2 response, which is generally protective for helminths.

Overall, the study is well designed and competently executed. The statistical analyses appear adequate, although some of the analytical tools used are unfamiliar to me. Nevertheless, the reported significance seems consistent with the raw data presented. The discussion section, in my view, is somewhat verbose, and certain ideas could be expressed more concisely to enhance readability.

One limitation of the study is its relatively modest novelty, as the authors themselves acknowledge that similar coinfection models (e.g., H. polygyrus / P. chabaudi) have been explored previously. However, I recognize that this consideration lies primarily within the editorial domain.

I would also encourage the authors to be more nuanced when describing "microparasitic" infections as inherently Th1-enhancing. The immune polarization in such infections can be dynamic and context-dependent. For instance, Leishmania spp. infections are initially associated with Th1 responses but often shift toward Th2 dominance during chronic stages. Greater precision in such immunological characterizations would strengthen the manuscript’s interpretative depth.

From a formal standpoint, I find that the inclusion of detailed statistical information directly within the main text—especially in the Results section—detracts from readability. It would improve clarity if these details were moved to figure legends or supplementary materials.

In conclusion, this is a well-executed and appropriately analyzed study that, while not groundbreaking, meaningfully contributes to the understanding of how coinfection influences helminth pathobiology. The findings have potential relevance for public health in regions where such infections are endemic, assuming the mouse model’s applicability to human coinfection can be reasonably extended. I therefore recommend publication after minor stylistic revisions.

Minor comments:

Line 96–97: Consider rephrasing to: "at the acute phase of infection, primarily."

Line 149: I am not an expert in helminth handling, but is distilled water standard practice for larval processing, and is it known to be non-harmful to larvae?

Reviewer #2: This is a well-designed, mechanistically informative study investigating how Plasmodium yoelii coinfection alters Heligmosomoides polygyrus fecundity and persistence through IL-13–mediated immune modulation. The experimental design is strong, the findings are clear, and the topic is a good fit for this journal.

The manuscript’s primary weakness is the incomplete mechanistic coverage—most immune readouts are systemic, and there are no intestinal (local) data confirming how IL-13 affects the worm’s niche. The statistical modeling (GLMMs) also requires clarification. These issues are addressable through revision.

With improved statistical transparency, moderated interpretation, and a clearer discussion of local immune effects, the paper would make a strong contribution to understanding helminth–malaria coinfection dynamics.

Reviewer #3: The manuscript reports how coinfection with Plasmodium alters the dynamics and fitness of the intestinal nematode H. polygyrus in C57BL/6 mice. Using controlled infection experiments, the authors show that malaria coinfection increases nematode egg excretion without increasing adult worm biomass, and that this effect can be induced even when H. polygyrus infection is already chronic. Overall, the study is well-designed and implemented, and its findings are interesting, with clear relevance to co-infection biology and implications for STH epidemiology and control programs.

PLOS authors have the option to publish the peer review history of their article (what does this mean? ). If published, this will include your full peer review and any attached files.). If published, this will include your full peer review and any attached files.

**Do you want your identity to be public for this peer review?** For information about this choice, including consent withdrawal, please see our For information about this choice, including consent withdrawal, please see our Privacy Policy ..

Reviewer #1: No

Reviewer #2: No

Reviewer #3: No

**Figure resubmission:**
---

## [Decision Letter · Decision Letter 1]

1 Feb 2026

Coinfection with malaria alters the fecundity and within-host persistence of an intestinal nematode

Dear Dr. Sorci,

Thank you for submitting your manuscript to PLOS Neglected Tropical Diseases. After careful consideration, we feel that it has merit but does not fully meet PLOS Neglected Tropical Diseases's publication criteria as it currently stands (Minor Revision). Therefore, we invite you to submit a revised version of the manuscript that addresses the points raised during the review process.

* A letter that responds to each point raised by the editor and reviewer(s). You should upload this letter as a separate file labeled 'Response to Reviewers '. This file does not need to include responses to any formatting updates and technical items listed in the 'Journal Requirements' section below.'. This file does not need to include responses to any formatting updates and technical items listed in the 'Journal Requirements' section below.

* A marked-up copy of your manuscript that highlights changes made to the original version. You should upload this as a separate file labeled 'Revised Manuscript with Track Changes '.'.

* An unmarked version of your revised paper without tracked changes. You should upload this as a separate file labeled 'Manuscript '.'.

We look forward to receiving your revised manuscript.

Kind regards,

Chao Yan

Academic Editor

Gabriel Rinaldi

Section Editor

Shaden Kamhawi

co-Editor-in-Chief

Paul Brindley

co-Editor-in-Chief

**Additional Editor Comments:**

There are still some concerns of the reviewers about the manuscript, please respond and revise them one point to one point.

**Journal Requirements:**

1) Please provide an Author Summary. This should appear in your manuscript between the Abstract (if applicable) and the Introduction, and should be 150-200 words long. The aim should be to make your findings accessible to a wide audience that includes both scientists and non-scientists. Sample summaries can be found on our website under Submission Guidelines:

**Reviewers' comments:**

Reviewer's Responses to Questions

**Key Review Criteria Required for Acceptance?**

**Methods**

-Are the objectives of the study clearly articulated with a clear testable hypothesis stated?

-Is the study design appropriate to address the stated objectives?

-Is the population clearly described and appropriate for the hypothesis being tested?

-Is the sample size sufficient to ensure adequate power to address the hypothesis being tested?

-Were correct statistical analysis used to support conclusions?

-Are there concerns about ethical or regulatory requirements being met?

Reviewer #2: The authors have provided a thorough and satisfactory response to all points raised in the initial review. They have improved the statistical transparency, refined the terminology regarding "fitness," and appropriately contextualized the study's limitations regarding local intestinal immunity and human translation.

Reviewer #4: If possible, I recommend consulting a statistician, as I am not certain that ANOVA is the most appropriate test for all of the experimental results presented. In particular, when comparisons involve only two groups (e.g., infected vs. non-infected, or co-infected vs. single-infected), alternative statistical tests may be more suitable.

**Results**

-Does the analysis presented match the analysis plan?

-Are the results clearly and completely presented?

-Are the figures (Tables, Images) of sufficient quality for clarity?

Reviewer #2: Yes, the revised paper meet these criteria.

Reviewer #4: (No Response)

**Conclusions**

-Are the conclusions supported by the data presented?

-Are the limitations of analysis clearly described?

-Do the authors discuss how these data can be helpful to advance our understanding of the topic under study?

-Is public health relevance addressed?

Reviewer #2: Yes, the revised paper meets these criteria.

Reviewer #4: (No Response)

**Editorial and Data Presentation Modifications?**

Reviewer #2: I had a few small suggestions based on a reading of the revised paper:

The methods state that only female mice were used. Given the known sex differences in parasitology (males often having higher worm burdens), the authors should justify this choice or discuss it as a limitation for translation.

For the longitudinal parasitemia data (Fig 1), the authors appear to use repeated t-tests. Please use a Repeated Measures ANOVA or a Mixed Effects Model to account for the temporal correlation of data within the same mouse.

The discussion should expand on the implications for malaria vaccination. If helminths suppress Th1 responses, would this render a malaria vaccine ineffective in a coinfected population? This would strengthen the "Significance" section.

Reviewer #4: (No Response)

**Summary and General Comments**

Reviewer #2: This manuscript provides clear evidence of immunological antagonism between a nematode and malaria. In it's current form, the paper is largely a descriptive study. In the future, the authors should seek to prove the causal role of the regulatory pathway (IL-10/Treg) and address the clinical outcome of anemia.

Reviewer #4: Minor revisions:

Author Summary:

Line 54: Please replace malaria with Plasmodium.

Line 57: Please replace nematode with soil-transmitted helminths.

Discussion:

Line 532: Please replace malaria with Plasmodium.

I also suggest moving the first paragraph of the Discussion (lines 532–534) to another section, as it reads more like a conclusion. A possible improvement would be to start the Discussion with the paragraph beginning at line 535.

PLOS authors have the option to publish the peer review history of their article (what does this mean? ). If published, this will include your full peer review and any attached files.). If published, this will include your full peer review and any attached files.

**Do you want your identity to be public for this peer review?** For information about this choice, including consent withdrawal, please see our For information about this choice, including consent withdrawal, please see our Privacy Policy ..

Reviewer #2: No

Reviewer #4: **Yes:** Lopes-Torres EJLopes-Torres EJ

**Figure resubmission:**
---

## [Editor Report · Decision Letter 2]

3 Mar 2026

Dear Dr. Sorci,

We are pleased to inform you that your manuscript 'Coinfection with malaria alters the fecundity and within-host persistence of an intestinal nematode' has been provisionally accepted for publication in PLOS Neglected Tropical Diseases.

Best regards,

Chao Yan

Academic Editor

Gabriel Rinaldi

Section Editor

Shaden Kamhawi

co-Editor-in-Chief

Paul Brindley

co-Editor-in-Chief

---

## [Editor Report · Acceptance letter]

Dear Dr. Sorci,

We are delighted to inform you that your manuscript, "Coinfection with malaria alters the fecundity and within-host persistence of an intestinal nematode," has been formally accepted for publication in PLOS Neglected Tropical Diseases.

Best regards,

Shaden Kamhawi

co-Editor-in-Chief

Paul Brindley

co-Editor-in-Chief
